# Development of Neural Networks to Study Flow Behavior of Medium Carbon Microalloyed Steel during Hot Forming

**Anas Al Omar** *, **Pau Català** *, **Jose Ignacio Alcelay** and **Esteban Peña**

Department of Mechanical Engineering, Universitat Politècnica de Catalunya, Escola Politècnica Superior d'Enginyeria de Manresa (EPSEM), Av. de les Bases de Manresa 61-73, 08242 Manresa, Spain; inaki.alcelay@upc.edu (J.I.A.); esteban.pena@upc.edu (E.P.)

* Correspondence: anas.al.omar@upc.edu (A.A.O.); pau.catala@upc.edu (P.C.)

**Abstract:** In the present article, the application of an artificial neural network (ANN) model whose function is the development of plastic instability maps of a medium carbon microalloyed steel during the hot forming process is studied. Secondly, we proceed to create another ANN capable of providing the recrystallized grain size in the steady state resulting from forming deformation. We start from the experimental data of a medium carbon microalloyed steel obtained by hot compression tests with strain rates that vary between $10^{-4}\ \text{s}^{-1}$ and $3\ \text{s}^{-1}$ and in a range of temperatures between 900 °C and 1150 °C. These experimental data are used to train the proposed ANN and obtain flow curves. Finally, the processing maps are developed by applying the dynamic materials model (DMM), according to which the safe hot forming domains and the plastic instability domains of the studied material are delineated. The comparison between the ANN and the experimental maps is carried out. It is ascertained that the optimal regions of forging in the ANN maps coincide with those obtained in the experimental maps. In addition, a study of the influence of the microstructure on the behavior of the studied steel during hot forming is carried out.

**Keywords:** artificial neural network; dynamic material model; processing maps; flow behavior; medium carbon microalloyed steel





## 1. Introduction

For various decades, medium carbon microalloyed steels have been used in the manufacturing of industrial parts. These steels are basically C-Mn steels with additions of conventional microalloying elements such as aluminum (Al), vanadium (V), titanium (Ti) or niobium (Nb). These microalloying elements contribute to improving the mechanical properties of these steels by both grain size refinement and precipitation hardening during component cooling after hot forming, without the need to use heat treatment processes, which is interesting for several industrial applications [1], such as automotive [2] or railway transportation applications [3]. Unfortunately, due to the complex relationships involved in hot forming process modelling and the non-linearity of their flow behavior, there are very few works in the literature using artificial neural networks (ANNs) to predict the flow behavior of medium carbon microalloyed steels during the hot forming process. ANN algorithms are recognized as among the most powerful algorithms in alloy design included in Machine Learning (ML) techniques [4]. Shina et al. [5] stated that methods to minimize experiments to predict mechanical properties based on ML techniques are interesting for the industry, so they developed an ANN for microalloyed steel for the hot rolling–direct strip combined process. Pan et al. [6] compared five ML algorithms (which included ANNs) for a Ni-Cr-Mo low-alloy steel to predict flow stress. Ghazani developed an efficient ANN to predict the flow behavior of Ti-Nb microalloyed steel during hot torsion deformation [2].

The dynamic materials model (DMM) [7,8] is a method capable of characterizing hot forming processes by analyzing and optimizing the hot formability of numerous

materials. In the DMM, to characterize the flow behavior of materials, the potential constitutive law ($\sigma = K\dot{\varepsilon}^m$) is used, both in low- and high-stress domains. But, in the works published by Narayana Murthy et al. [9–11], it has been shown that the potential constitutive law cannot be used indiscriminately in low- and high-stress domains. It is well known that it can be used in the analysis of steady-state stresses under low-stress forming conditions. For this reason, Narayana Murty et al. [9–11] reanalyzed the DMM and proposed another methodology based on obtaining the energy dissipation efficiency directly through numerical integration. This methodology can be considered a variant of the DMM (VDMM). In order to obtain the optimal hot forming zones and, consequently, safe parts without microstructural defects (cracks, fissures, cavities, etc.), it is necessary to skillfully and accurately apply the DMM and VDMM methodologies, analyzing the evolution of flow stress as a function of temperature and strain rate.

Another important parameter is the initial grain size because the microstructure of a deformed material and its flow curves are very sensitive to this parameter. Numerous studies have confirmed that the final microstructure resulting from microstructural evolution during the hot forming process has a decisive effect on the final properties of the conformed material. Thus, in the present work, in addition to temperature, strain and strain rate, initial grain size is introduced as input data and a control parameter for the development of an artificial neural network to characterize the hot forming of the studied steel.

Artificial neural networks (ANNs) are hardware and/or software constructions that take input information and transform it into an output, generally by applying non-linear operations. They are simplified calculation models inspired by biological neural networks of the human brain. An ANN consists of a number of interconnected processing elements (perceptrons) called neurons. Neurons are organized in different layers, in ways similar to those in the human brain [4,12]. The processing elements of the neural network are distributed by layers. In each layer are the set of elements that are on the same level of the structure. There are input, intermediate and output layers. When applying models based on ANNs, the following three phases are required:

1. Training phase, also called learning phase, used to adjust the parameters of each model trained;
2. Validation phase, used to check how well each model adjusts with data unseen before for the models and to tune the corresponding hyperparameters. A set of test data is used to provide the unbiased evaluation of the final fit of a model on the training data set.
3. Testing or test phase, used to quantify the ability of each trained model to predict feasible output data. The chosen model is the one that has the best performance with the validation set of data.

Recently, new works that use ANN as a robust tool capable of predicting the flow behavior of medium carbon steels have appeared. Tize Mha et al. [13,14], Quan et al. [15] Shekh, Kumar and Nath [16] and Ahmadi et al. [17], for different medium carbon steels with different alloy levels, concluded that ANN models represent the best alternative to generate flow curves by using experimental data to interpolate or extrapolate points. Pan et al. [6] concluded that the Random Committee algorithm could predict flow stress more effectively than ANNs for a Ni-Cr-Mo low-alloy steel.

When using ANN models, the discrepancies are basically in the treatment of data and the architecture of the network in issues such as input variables, range of values used for these input variables, number of hidden layers, number of neurons, transfer function, activation and the initialization of the weights. However, it is common to use yield stress as the output variable to use a Multi-Layer Perceptron network (MLP) based on Feed-Forward and with the backpropagation (BP) learning algorithm. The training phase stops when the mean square error is minimized. For the studied microalloyed steel, the proposed ANN architecture is similar to that in other works [13,14,16,17].

The novelties and originalities of great interest of this work are the following: (i) the contribution to understanding complex flow behavior and studying the dominant deformation mechanisms that control the microstructure, through grain size control, during hot forming for a non-studied medium carbon microalloyed steel; (ii) the implementation of an ANN capable to successfully predict the final grain size for the studied steel; (iii) the thermodynamic analysis of the deformation conditions that can lead to the appearance of plastic instabilities in the material.

## 2. Materials and Methods

### 2.1. Experimental Procedure

The commercial medium carbon microalloyed steel studied is intended for the forging sector for automotive components, and its chemical composition is represented in Table 1.

**Table 1.** Chemical composition of the studied medium carbon microalloyed steel.

| %C | %Mn | %Si | %P | %S | %V | %Ti | %Al | Nppm |
|----|-----|-----|----|----|----|-----|-----|------|
| 0.29 | 1.19 | 0.19 | 0.012 | 0.025 | 0.09 | 0.002 | 0.011 | 131 |

The experimental method used to obtain the experimental data is the same used in previous works by Alcelay, Al Omar and Prado [18] and Al Omar [19]. For the hot compression test used to study the flow behavior of the studied steel, two testing machines were employed, depending on the required true strain rate: an electromechanical machine for $10^{-4}\,\text{s}^{-1} \leq \dot{\varepsilon} \leq 0.1\,\text{s}^{-1}$ and a servohydraulic machine for $\dot{\varepsilon} = 1\,\text{s}^{-1}$ and $\dot{\varepsilon} = 3\,\text{s}^{-1}$. The test temperatures were varied from 900 °C to 1150 °C. Prior to the hot compression tests, the cylindrical specimens (which were 11.4 mm in height and 7.6 mm in diameter) were austenitized for 30 min directly at the test temperatures. Therefore, the initial grain size at each deformation temperature was different. Moreover, since the number of elements put into solution or precipitated was different at each test temperature, different alloys were obtained at each temperature. It must be noted that the aim of this austenitizing treatment is to determine if the coarsening of precipitates affects the flow behavior, by comparing the results with results previously published by the authors where specimens were austenitized for 5 min at the test temperature [18,19].

The austenitizing treatments were carried out in a tube furnace capable of reaching 1500 °C and in a protective atmosphere of argon to avoid possible decarburization. The specimens were introduced directly into the furnace when it reached the test temperature. The holding time started when the furnace temperature was stabilized. Once the hot compression test was finished, specimens were quenched immediately in water to maintain the deformed microstructure. The metallographic preparation of the specimens was carried out for a subsequent measurement of the austenitic grain size. After polishing, the specimens were subjected to chemical attack in order to reveal the austenitic grain boundaries, and the grain size was measured by optical microscopy by using a computer-based image analyzer.

In order to avoid the friction effects at the die–specimen contact surface, boron nitride was used as lubricant only for strain rates of $1\,\text{s}^{-1}$ and $3\,\text{s}^{-1}$. For low-strain-rate tests (i.e., $\leq 0.1\,\text{s}^{-1}$), lubrication was finally not used, since it was verified that its effect was negligible. However, at the strain rate of $0.1\,\text{s}^{-1}$ (maximum strain rate reached in the electromechanical universal testing machine) and at low temperatures (900 °C and 950 °C), it was detected that the deformed specimens had a small barrel shape due to frictional forces, resulting in a small increase in flow stresses at higher degrees of deformation (from 0.7). Figure 1 shows the initial microstructure of the studied steel austenitized for 30 min at test temperatures of 1050 °C and 1150 °C [19].

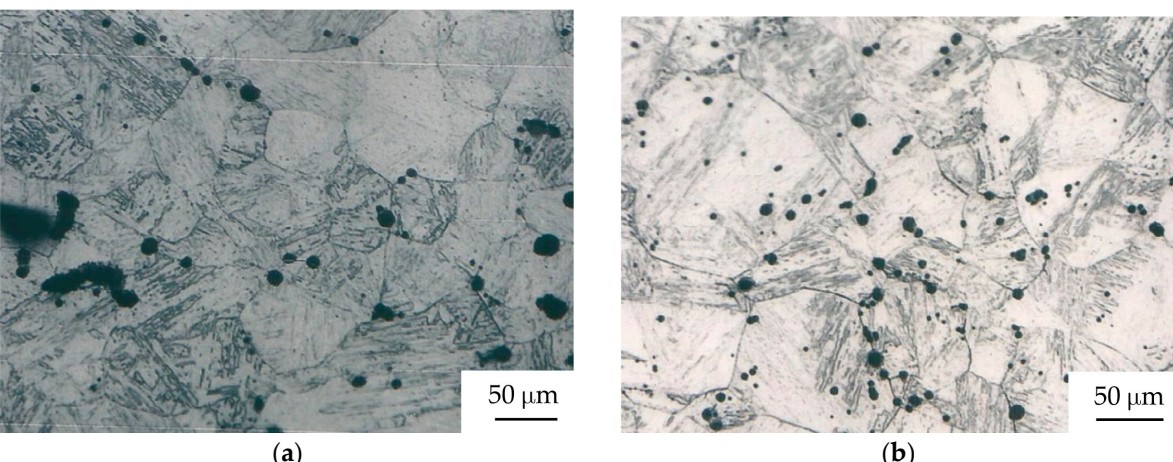

(**a**)                                                                                                      (**b**)

**Figure 1.** Initial microstructure of studied steel austenitized for 30 min: (**a**) 1050 °C; (**b**) 1150 °C.

Table 2 contains the selected experimental flow curves, for different temperatures and strain rates, used for the construction of the ANN.

**Table 2.** Selected experimental flow curves for different temperatures and strain rates used for the construction of the ANN of the studied steel.

| *T* (°C) | Strain Rates $\dot{\varepsilon}$ (s$^{-1}$) |
|---|---|
| 900 | **10$^{-4}$**, 5 × 10$^{-4}$, **10$^{-3}$**, 2 × 10$^{-3}$, 3 × 10$^{-3}$, 5 × 10$^{-3}$, **10$^{-2}$**, 2 × 10$^{-2}$, 3 × 10$^{-2}$, 5 × 10$^{-2}$, 7 × 10$^{-2}$ |
| 950 | 2 × 10$^{-4}$, 5 × 10$^{-4}$, **10$^{-3}$**, 2 × 10$^{-3}$, 5 × 10$^{-3}$, **10$^{-2}$**, 2 × 10$^{-2}$, 5 × 10$^{-2}$, **10$^{-1}$**, **1**, **3** |
| 1000 | **10$^{-4}$**, 2 × 10$^{-4}$, 5 × 10$^{-4}$, 6 × 10$^{-4}$, 7 × 10$^{-4}$, **10$^{-3}$**, 2 × 10$^{-3}$, 5 × 10$^{-3}$, **10$^{-2}$**, 2 × 10$^{-2}$, 5 × 10$^{-2}$, **10$^{-1}$**, **1**, **3** |
| 1050 | **10$^{-4}$**, 5 × 10$^{-4}$, **10$^{-3}$**, 5 × 10$^{-3}$, **10$^{-2}$**, 2 × 10$^{-2}$, 5 × 10$^{-2}$, **10$^{-1}$**, **1**, **3** |
| 1100 | **10$^{-4}$**, 2 × 10$^{-4}$, 3 × 10$^{-4}$, 5 × 10$^{-4}$, **10$^{-3}$**, 3 × 10$^{-3}$, **10$^{-2}$**, 2 × 10$^{-2}$, 3 × 10$^{-2}$, **10$^{-1}$**, **1**, **3** |
| 1150 | **1**, **3** |

### 2.2. Neural Network Model to Obtain Processing Maps

#### 2.2.1. Data Preparation

The experimental raw data were previously treated to facilitate the development of an efficient ANN for the different phases. A large number and variety of experimental data were available (see Table 2) which required us to unify the selection criteria thereof. The input data (inputs) and the output data (outputs) were treated by adopting the following rule: the flow curves used were those that corresponded to the main strain rates, that is, 10$^{-4}$ s$^{-1}$, 10$^{-3}$ s$^{-1}$, 10$^{-2}$ s$^{-1}$, 10$^{-1}$ s$^{-1}$, 1 s$^{-1}$ and 3 s$^{-1}$ (in bold in Table 2), since they were curves whose experimental data were repeated at most of the temperatures studied. This was applied during the training and validation phase of the ANN. The intermediate strain rates were used for the testing phase. In addition, since the ANN requires that both the input and output data be between 0 and 1 and in order to accelerate convergence in the training phase, we are interested in the data having small values to ensure the settlement of the network into a stable solution. First, for *T*, the logarithms of the experimental values are used; then, they are normalized between 0 and 1, as similarly performed by G. Quan et al. [15]. For *T* and $\sigma$, the following equation is applied:

$$Z' = \frac{Z - 0.95\,Z_{\min}}{1.05\,Z_{\max} - 0.95\,Z_{\min}} \tag{1}$$

where $Z$ are the experimental data ($T$ and $\sigma$) in logarithms and $Z'$ is the normalized value of $Z$, which has a maximum and a minimum value given by $Z_{max}$ and $Z_{min}$, respectively. However, Equation (1) cannot be used to normalize the values due to the values obtained being too small. To normalize the values, the following equation is used:

$$\dot{\varepsilon}' = \frac{5 + \log \dot{\varepsilon} - 0.95(5 + \log \dot{\varepsilon}_{min})}{1.05(5 + \log \dot{\varepsilon}_{max}) - 0.95(5 + \log \dot{\varepsilon}_{min})} \tag{2}$$

In Equation (2), the value 5 is used so that all normalized values are positive. This value will depend on the minimum deformation speed that we have in the experimental data of the studied microalloyed steel. It is not necessary to normalize the deformation $\varepsilon$ since its values are between 0 and 1.

The experimental data used to train and validate the network are presented in Table 3. From the 33 experimental data base flow curves, 22 random flow curves (66.67%) were used to train and 5 flow curves to validate (15.15%) the ANN model. For the testing phase, 6 flow curves (18.18%) were used.

**Table 3.** Experimental data, obtained from experimental flow curves of the studied steel, used for ANN validation.

| $\dot{\varepsilon}$ (s$^{-1}$) | Training | Validation |
|---|---|---|
| $10^{-4}$ | 900 °C, 1000 °C, 1100 °C | 1050 °C |
| $10^{-3}$ | 900 °C, 1000 °C, 1050 °C, 1100 °C | 950 °C |
| $10^{-2}$ | 900 °C, 950 °C, 1050 °C, 1100 °C | 1000 °C |
| $10^{-1}$ | 950 °C, 1000 °C, 1100 °C | - |
| 1 | 950 °C, 1050 °C, 1100 °C, 1150 °C | 1000 °C |
| 3 | 950 °C, 1000 °C, 1050 °C, 1150 °C | 1100 °C |

The flow stress results obtained by validating or testing the network make it possible to define its reliability. These values are those that correspond to the temperatures and strain rates that have not been used for the training and validation phases. Table 4 represents the data used to verify the ANN testing phase.

**Table 4.** Experimental data of the studied steel used for the ANN testing phase.

| $\dot{\varepsilon}$ (s$^{-1}$) | Testing |
|---|---|
| $3 \times 10^{-3}$ | 900 °C |
| $5 \times 10^{-3}$ | 950 °C |
| $5 \times 10^{-4}$ | 950 °C, 1000 °C, 1050 °C, 1100 °C |

### 2.2.2. Neural Network Model

The network model used is the Multi-Layer Perceptron (MLP) based on Feed-Forward and backpropagation (BP), used as a supervised learning algorithm. For training, the Levenberg–Marquardt algorithm (trainlm) based on backpropagation is applied. The network itself determines the weights randomly. In order to choose the best ANN architecture, tan-sigmoid and log-sigmoid were compared as a transfer function. The log-sigmoid transfer function presented the best performance in the numerical experiments carried out during the training phase. Log-sigmoid was also used by Tize Mha et al. [13,14].

$$f(x) = \frac{1}{1 + e^{-x}} \tag{3}$$

The number of hidden layers, with their neurons, and the minimum square error (MSE) determine the highest efficiency in the results for each steel. It is obtained by evaluating standard statistical indices from the results achieved during the training and validation phases. The final training tests are carried out.

The architecture of the network adopted is made up of three layers, an input layer made up of three elements ($T$, $\varepsilon$ and $\dot{\varepsilon}$), two hidden layers with twelve and nine neurons and voltage ($\sigma$) as the output layer. The architecture of the network, 3-12-9-1, is represented in Figure 2. The best results were obtained for an MSE of 0.00007. The training phase stopped after 224 iterations.

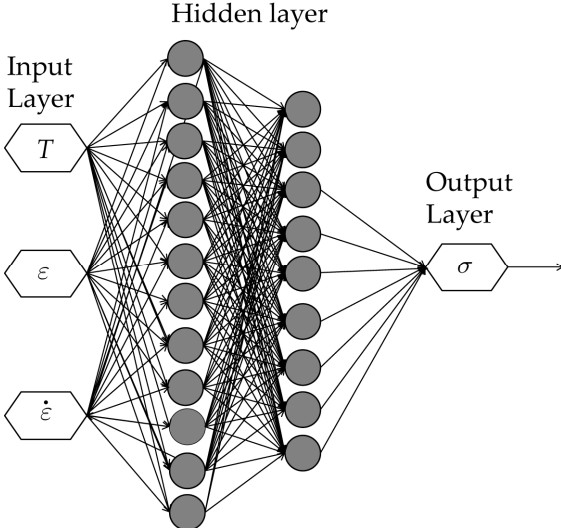

**Figure 2.** ANN model.

2.2.3. Standard Validation of Results of Neural Network

After the training phase of the network, the performance of the network is checked for the test curves, and the results obtained must be validated. A number of standard statistical indices are used to validate the feasibility of the results [20,21]. These methods are able to show the generalization capacity of network formation. It is quantified in terms of the correlation coefficient ($R$), the average relative absolute error ($e_{AARE}$), the square root of the mean square error ($RMSE$), the dispersion index ($SI$) and the relative error ($e_{rel}$; error between the experimental data and those obtained by the ANN). These parameters are defined as follows:

$$R = \frac{\sum\limits_{i=1}^{N} \left(E_i - \overline{E}\right)\left(P_i - \overline{P}\right)}{\sqrt{\left(\sum\limits_{i=1}^{N} \left(E_i - \overline{E}\right)^2 \sum\limits_{i=1}^{N} \left(P_i - \overline{P}\right)^2\right)}} \tag{4}$$

$$e_{AARE} = \frac{1}{N}\sum\limits_{i=1}^{N} \left| \frac{E_i - P_i}{E} \right| \cdot 100\% \tag{5}$$

$$RMSE = \sqrt{\frac{1}{N}\sum\limits_{i=1}^{N} (E_i - P_i)^2} \tag{6}$$

$$SI = \frac{RMSE}{\overline{E}} \tag{7}$$

$$e_{rel} = \left(\frac{E_i - P_i}{E_i}\right) \cdot 100\% \tag{8}$$

where $E_i$ is the experimental value and $P_i$ is the value obtained by using the neural network model; $\overline{E}$ and $\overline{P}$ are the mean values of $E$ and $P$, respectively; and $N$ is the total number of data used.

Small values (zero approximation) of $e_{AARE}$ and $RMSE$ mean a good correlation between predictive and experimental data. However, it is important to note that the highest values of coefficient $R$ (close to 1) should not always be interpreted as evidence of good predictive performance of the developed ANN model, because these values do not fully describe the relationship between the experimental values and those predicted [22,23]. Furthermore, to evaluate the ANN model, standard statistical indices are evaluated and presented in Table 5.

**Table 5.** ANN model performance in training, validation and testing phases.

|  | *R* | *RMSE* (%) | $e_{aare}$ (%) | **SI** |
|---|---|---|---|---|
| Training | 0.99781 | 2.8974 | 3.5978 | 0.0007 |
| Validation | 0.99377 | 4.1508 | 4.7632 | 0.0637 |
| Testing | 0.99059 | 3.5909 | 4.3209 | 0.0553 |

The error distributions of the flow stress of the neural network model in relation to the experimental data are shown in Figure 3.

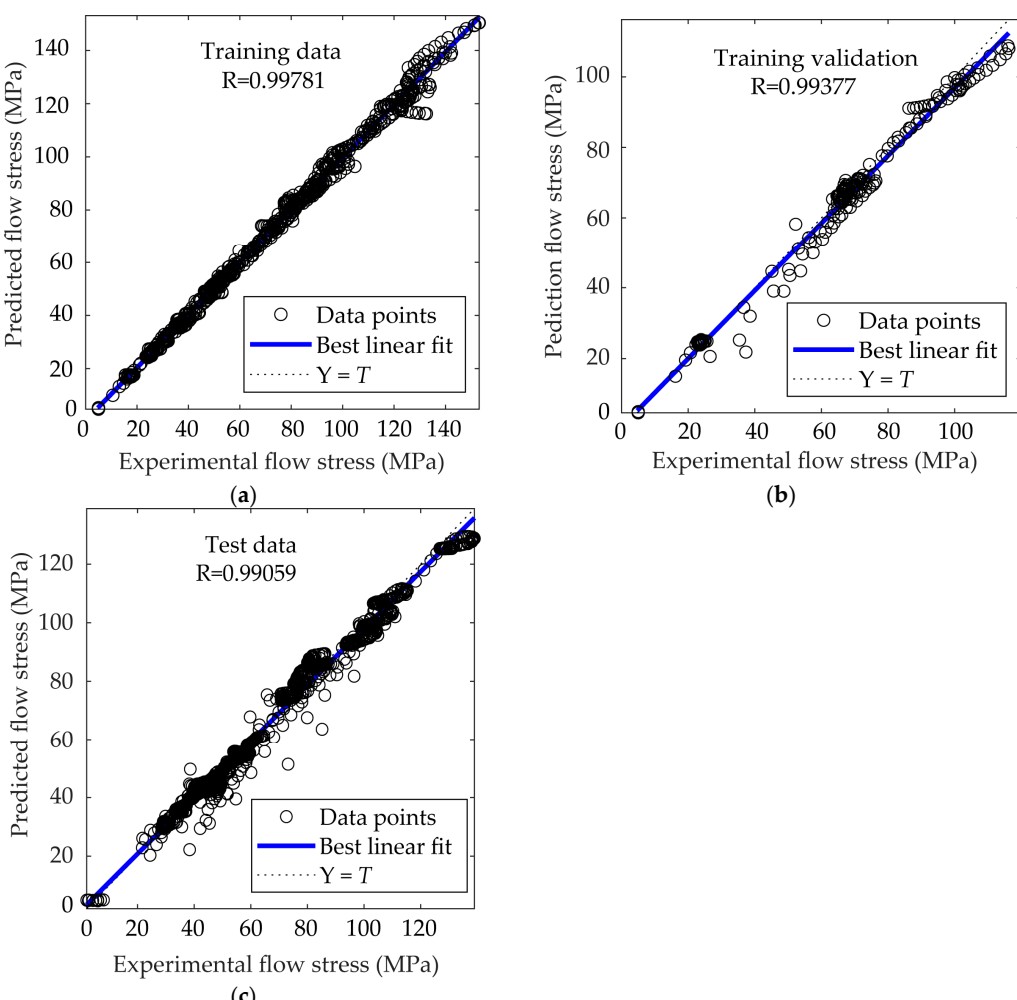

**Figure 3.** Comparison between the experimental flow stress and that obtained by the ANN for studied steel: (**a**) training; (**b**) validation; (**c**) testing.

As can be seen graphically in Figure 3, the values of the flow stress in the training, validation and testing phases of the ANN reveal that they are very similar to the experimental ones. Most of the points fall along the 45° line, and the correlation coefficients for training, validation and testing are 0.99781, 0.99377 and 0.99059, respectively. These *R* values are similar to those in others works to accept the goodness of the correlations between simulated and experimental data for other medium carbon microalloyed steels [24]. All the above results indicate that the ANN model has been successfully trained and can be applied to predict the flow stress behavior of the studied microalloyed steel.

To confirm the accuracy of ANN model performance, statistical analysis of the relative error is also used. The distributions of the relative error of the ANN model for the training, validation and testing phases are shown in Figure 4. This figure indicates that the predictions of the relative errors of the three data sets show a typical Gaussian distribution and show that are within 10% for more than 95% of the test data (a relative error of 5% is observed for more than 85% of the training data). Consequently, the good performance in the prediction of the proposed ANN model is confirmed.

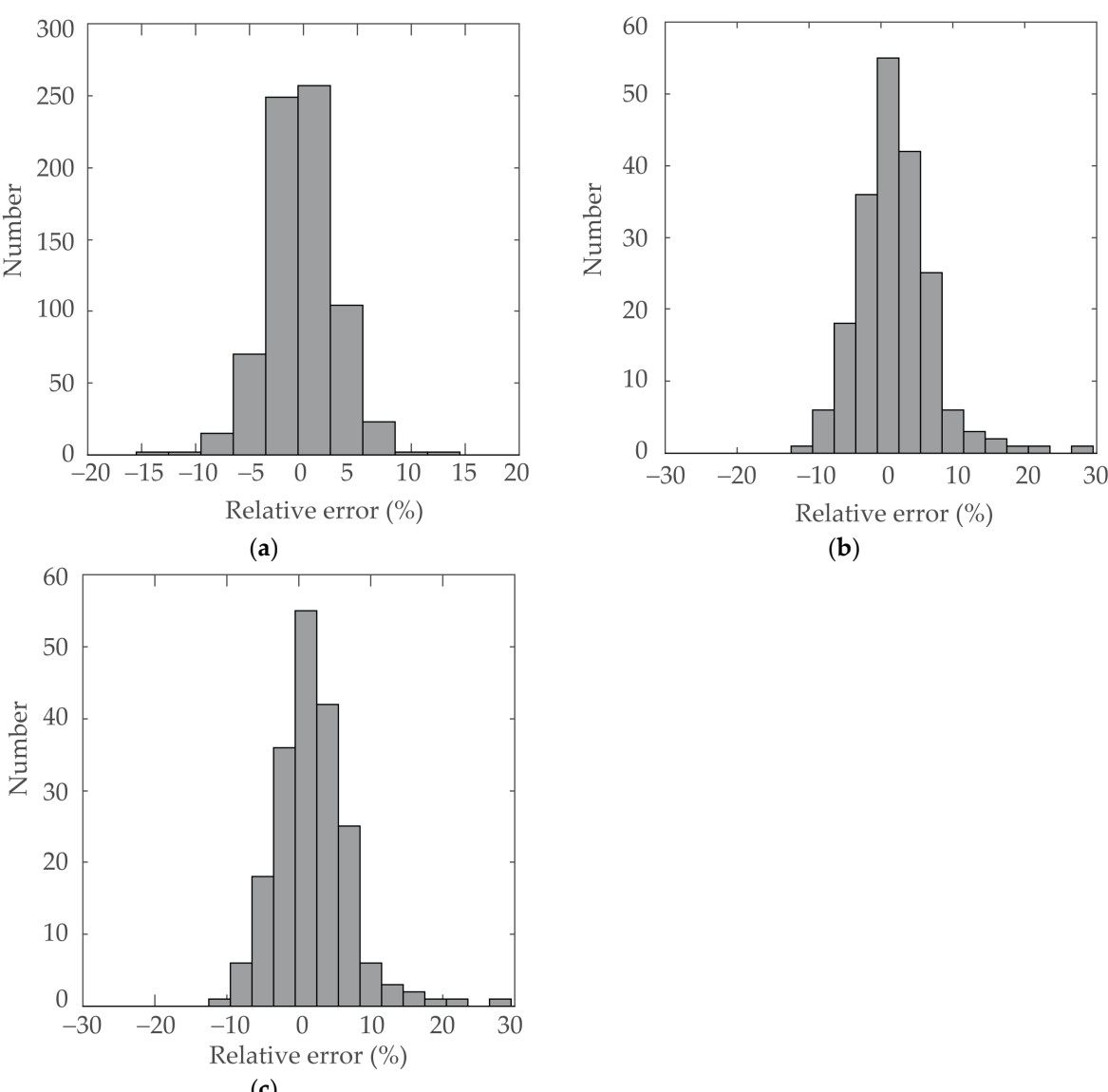

**Figure 4.** Statistical analysis of the prediction error of the ANN model for studied steel: (**a**) training; (**b**) validation; (**c**) testing.

## 3. Results

### 3.1. Application of Dynamic Model of Materials

3.1.1. Flow Curves

The flow curves of the studied steel are represented in Figures 5–8. The experimental flow curves are typical of materials that undergo dynamic recovery (DRV) and dynamic recrystallization (DRX), and the same can be said for the curves obtained through the ANN proposed model. The maximum peak stress ($\sigma_\text{p}$) and the peak strain ($\varepsilon_\text{p}$) increase with the strain rate for a given temperature. As the temperature increases, $\sigma_\text{p}$ and $\varepsilon_\text{p}$ decrease.

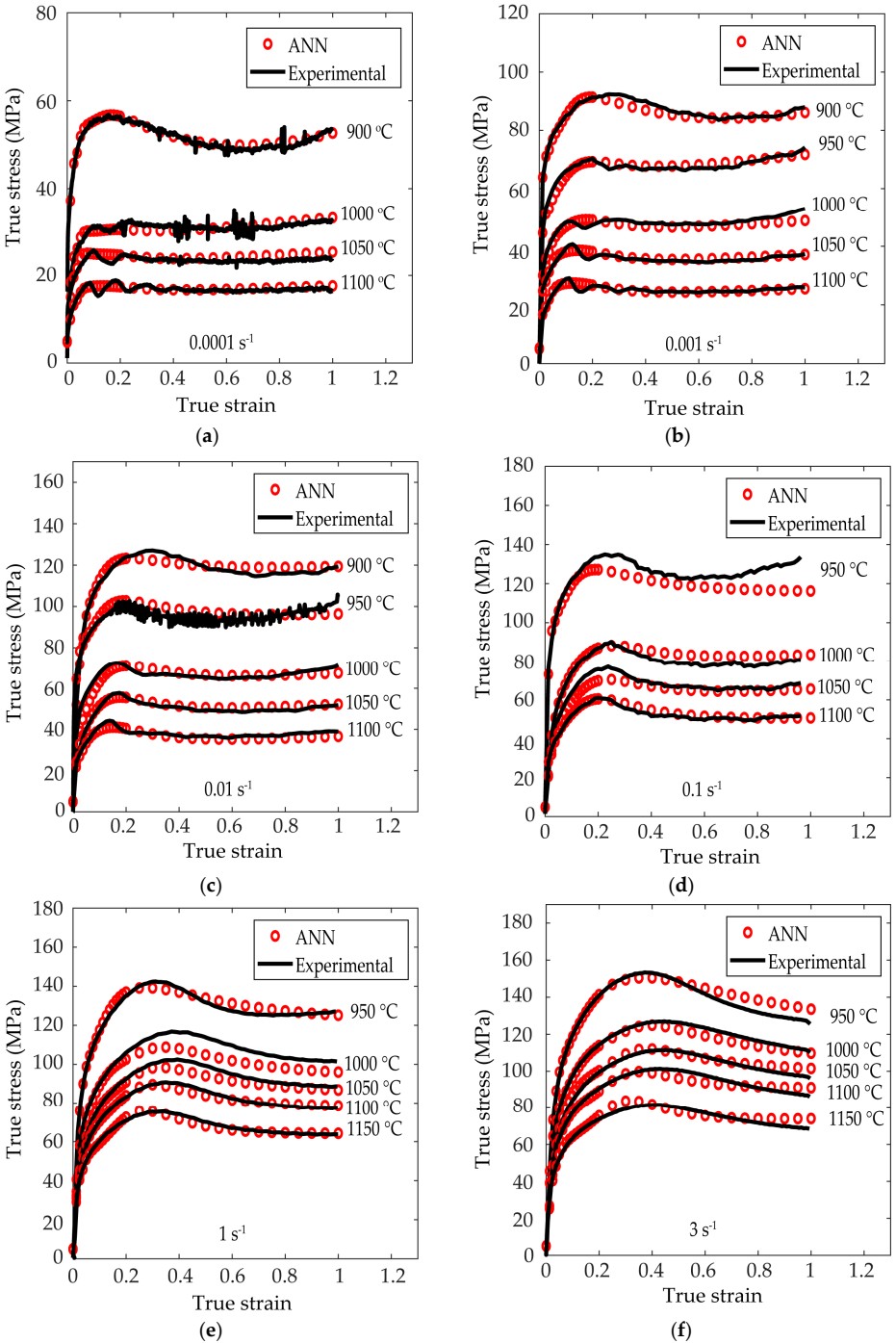

**Figure 5.** Flow curves obtained experimentally and by applying ANN for training and validation at different strain rates. (**a**) $\dot{\varepsilon}$ = 0.00001 s$^{-1}$; (**b**) $\dot{\varepsilon}$ = 0.001 s$^{-1}$; (**c**) $\dot{\varepsilon}$ = 0.01 s$^{-1}$; (**d**) $\dot{\varepsilon}$ = 0.1 s$^{-1}$; (**e**) $\dot{\varepsilon}$ = 1 s$^{-1}$; (**f**) $\dot{\varepsilon}$ = 3 s$^{-1}$.

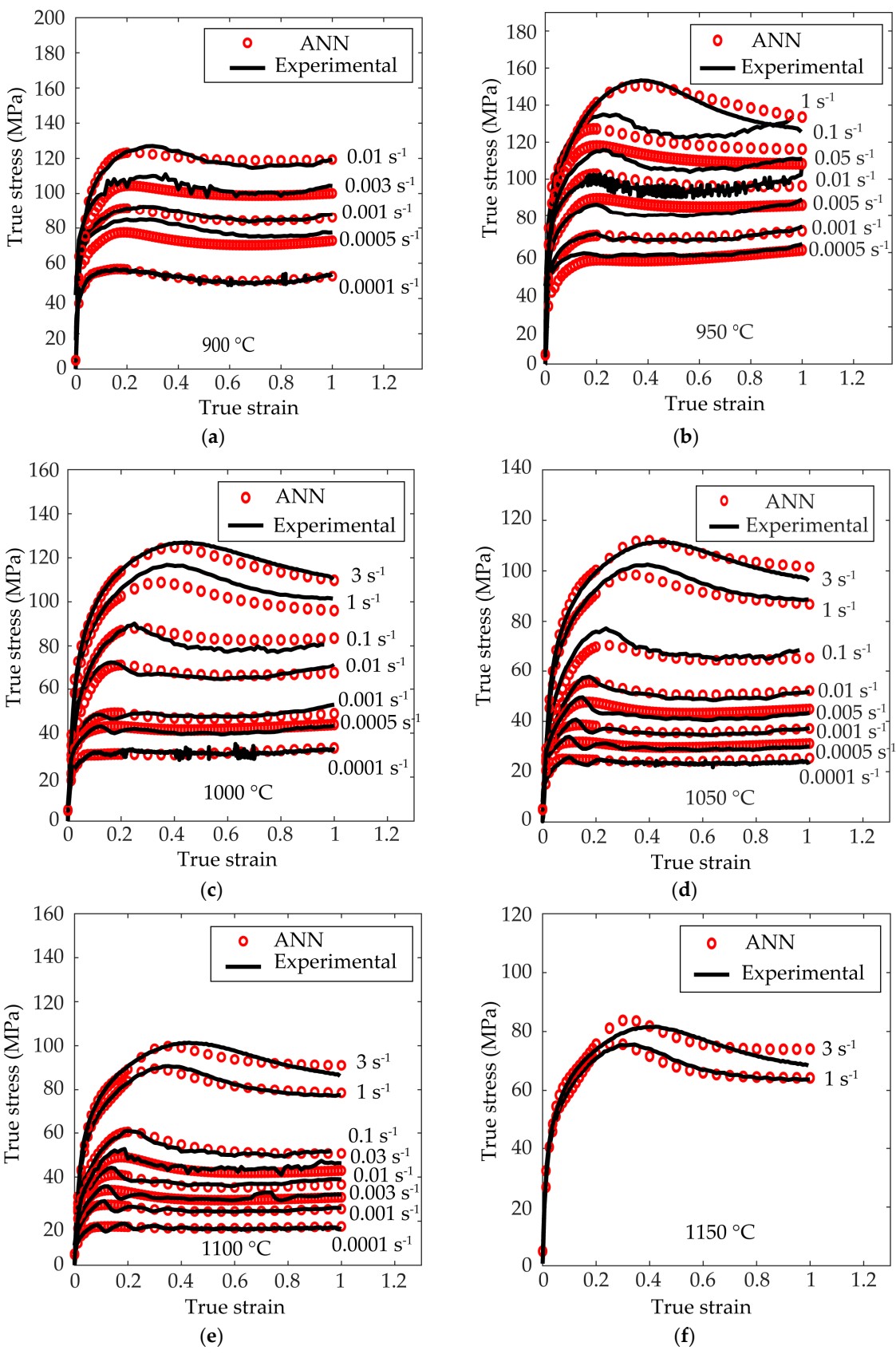

**Figure 6.** Flow curves obtained experimentally and by applying ANN for training, validation and testing phases at different temperatures. (**a**) *T* = 900 °C; (**b**) *T* = 950 °C; (**c**) *T* = 1000 °C; (**d**) *T* = 1050 °C; (**e**) *T* = 1100 °C; (**f**) *T* = 1150 °C.

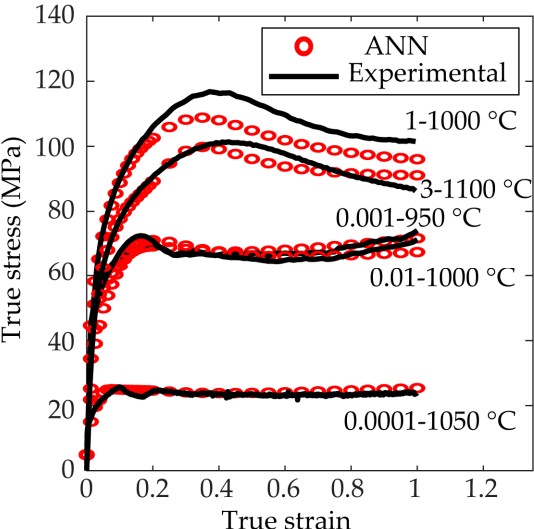

**Figure 7.** Flow curves obtained experimentally and by applying ANN after validation phase.

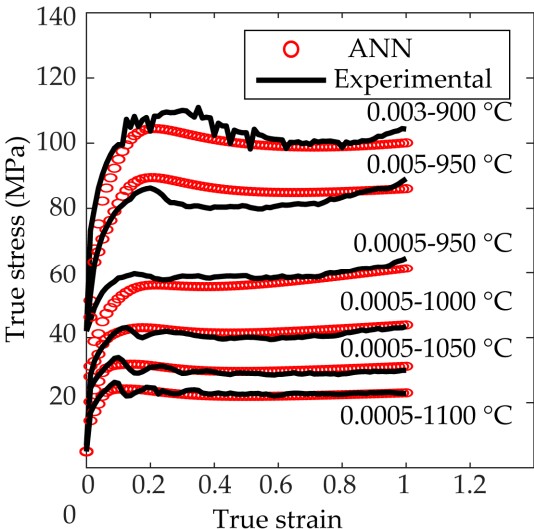

**Figure 8.** Flow curves obtained experimentally and by applying ANN after testing phase.

It can be seen that the proposed ANN model gives, under different deformation conditions, an accurate estimate of the flow curves and that the results are very similar.

Figure 5 represents the experimental flow curves (in black) and those obtained by the ANN after the training and validation phases (in red) for different temperatures. Figure 6 represents the experimental flow curves (in black) and those obtained by the ANN after the training, validation and testing phases (in red) for different strain rates. It can be seen in Figure 5d that the experimental flow curve at 950 °C and $\dot{\varepsilon} = 0.1$ s$^{-1}$ shows a small increase in stress from $\varepsilon = 0.7$ due to the frictional effects, as explained in Section 2.1. Figures 7 and 8 represent the flow curves for the temperatures and strain rates used for the ANN validation and testing phases, respectively.

### 3.1.2. Experimental Processing Maps Obtained by DMM and VDMM

In the research work carried out by Al Omar [19], the processing maps (energy dissipation efficiency maps and plastic instability maps) of the studied steel were obtained by using the DMM. The energy dissipation efficiency map reveals the existence of two domains characterized by maximum efficiency. The first domain occurs in the region of low temperatures and moderate strain rates (centered at approximately 900 °C and 0.0001 s$^{-1}$). In this case of low temperatures and intermediate strain rates, it is expected that DRV will

act; typical flow curves within this domain are characteristic of DRV. In the hot forming process, both DRX and DRV are considered beneficial mechanisms for the microstructure and consequently for the mechanical properties of the formed material, as they provide stable flow and improve the formability of the material [8,19]. The second domain appears centered at 1100 °C and 1 s$^{-1}$ with a maximum efficiency of approximately 31%; it is the domain of DRX. This correlation is confirmed by the flow curves obtained under different combinations of temperature and strain rate in this domain and clearly show continuous softening with single-peak behavior. For temperatures of 900–950 °C and high deformation speeds, a domain with low efficiency values appears and can be identified with zones that are not suitable for forming the material studied [8,19].

In relation to the plastic instability [19] map based on the DMM, as is known, the greater the negative magnitude of the plastic instability parameter ($\xi$), the greater the possibility of the appearance of some manifestation of plastic instability. Thus, in order to always form under stable flow conditions, domains of possible instabilities must be avoided during hot forming processes. The map represents two zones of stability; one for a temperature of 900 °C and a strain rate of 0.0001 s$^{-1}$ and the other for 1100 °C at 3 s$^{-1}$.

In Figure 9, the processing maps based on the VDMM are represented. These maps reveal a domain of plastic instability between 900 and 950 °C and high strain rates of 3 s$^{-1}$; this domain coincides with the domain found in the DMM maps. Despite the small difference found in the position of the stable domains, good agreement is observed between the different processing maps built based on the two models, the DMM and the VDMM.

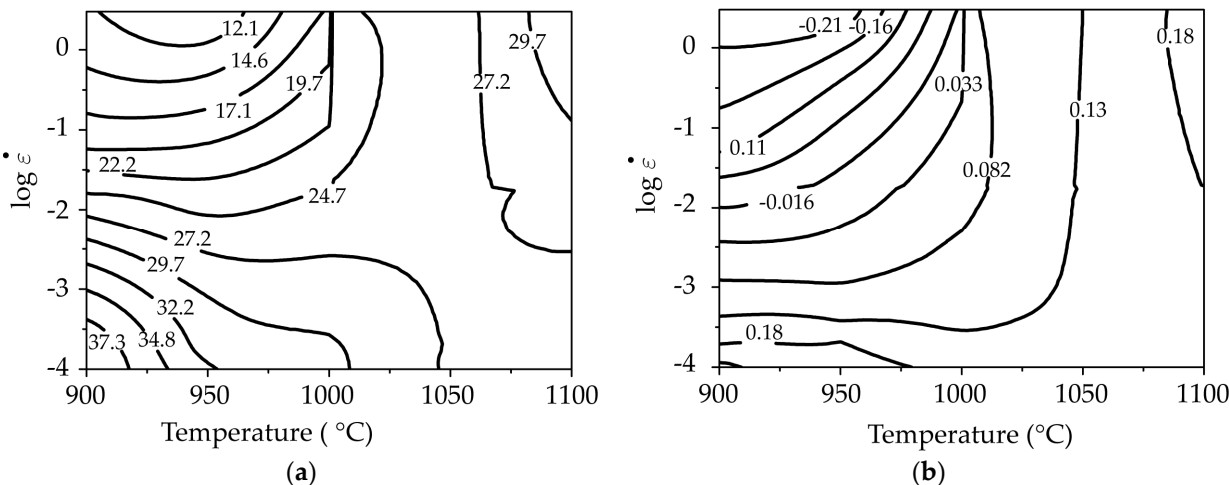

**Figure 9.** Experimental processing maps obtained by using VDMM: (**a**) isoefficiency map; (**b**) plastic instability map.

### 3.2. Application of ANN to Develop Processing Maps

To check the reliability of the ANN proposed in this work, processing maps were built based on the DMM (Figure 10) and on the VDMM (Figure 11) by using the flow curves obtained by the ANN. These maps coincide with those obtained through the experimental stress–strain curves obtained by applying the DMM and VDMM.

The DRX domain appears at 1100 °C and 3 s$^{-1}$ with an approximate efficiency of 32%, and another DRV domain appears at low temperatures and strain rates, centered approximately at 900 °C and 0.0001 s$^{-1}$. It is well known that both DRX and DRV domains are considered safe and are preferred for hot forming processes, because these mechanisms improve workability by reconstituting microstructure and decreasing the flow stress. In addition, an instability domain appears at low temperatures and high strain rates (900 °C and 3 s$^{-1}$) with very low efficiency, and it would be convenient not to conform under these conditions. These results have been confirmed by microstructural examinations in previous research studies carried out by the authors [19].

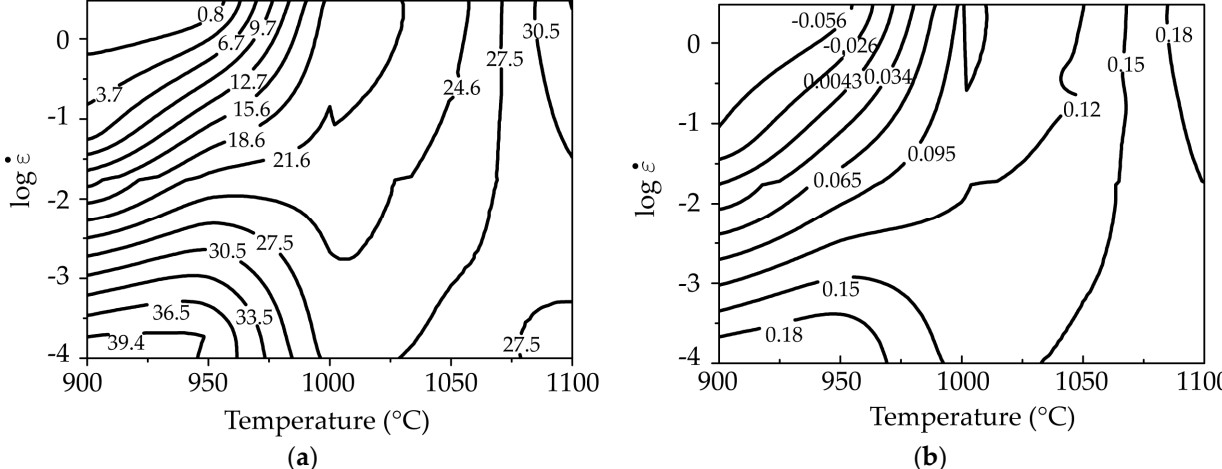

**Figure 10.** Processing maps obtained through ANN and based on DMM for $\varepsilon$ = 0.6: (**a**) isoefficiency map; (**b**) plastic instability map.

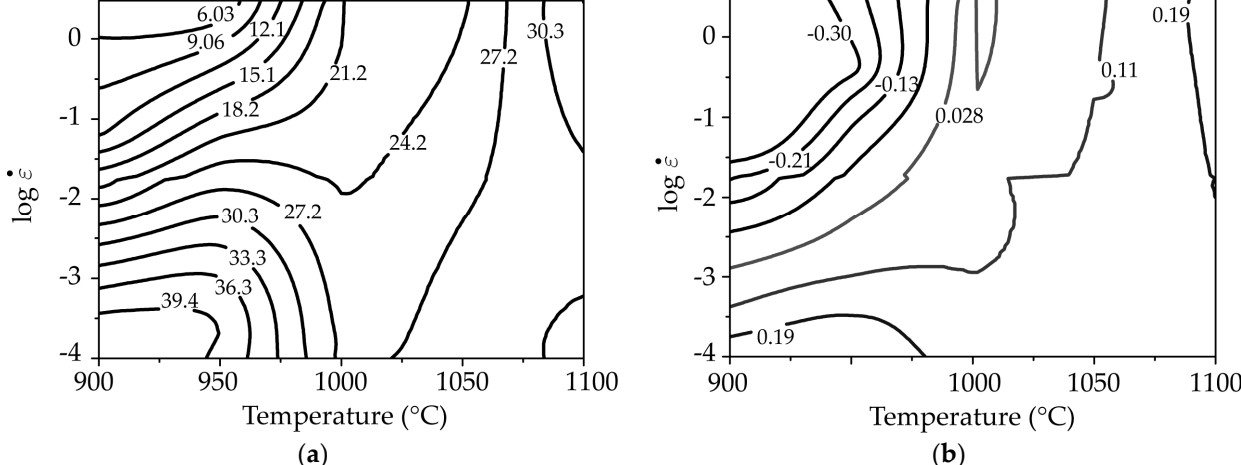

**Figure 11.** Processing maps obtained through ANN and based on the VDMM for $\varepsilon$ = 0.6: (**a**) isoefficiency map; (**b**) plastic instability map.

The similarity between the experimental processing maps and those obtained by using the ANN suggest, for future research, that the ANN developed in this work can be implemented in commercial finite element code to more efficiently simulate the hot flow behavior of medium carbon microalloyed steels.

### 3.3. ANN to Obtain Recrystallized Grain Size

#### 3.3.1. Experimental Data on Grain Size

The first fourth columns of Table 6 represents the experimental data of the initial grain size and the final grain size at the deformation of 0.6, corresponding to the stable state, for different temperatures and strain rates.

**Table 6.** Experimental initial and final grain sizes for different temperatures and strain rates and data obtained through ANN in training.

| $T$ (°C) | $\dot{\varepsilon}$ (s$^{-1}$) | $G_{s.INITIAL}$ (µm) | $G_{s.FINAL}$ (µm) | $ANN$ (µm) |
|---|---|---|---|---|
| 900 | 0.0001 | 11.12 | 22.2557 | 21.6098 |
| 900 | 0.0005 | 11.12 | 14.81 | 15.4256 |
| 900 | 0.0020 | 11.12 | 12.9680 | 11.1386 |

**Table 6.** *Cont.*

| $T$ (°C) | $\dot{\varepsilon}$ (s$^{-1}$) | $G_{s.INITIAL}$ (µm) | $G_{s.FINAL}$ (µm) | $ANN$ (µm) |
|---|---|---|---|---|
| 900 | 0.0100 | 11.12 | 11.3415 | - |
| 900 | 0.0500 | 11.12 | 9.799 | 9.8309 |
| 900 | 0.3000 | 11.12 | 8.3298 | 9.7824 |
| 950 | 0.0002 | 12.63 | 33.9160 | 33.4308 |
| 950 | 0.0005 | 12.63 | 26.6486 | 26.5033 |
| 950 | 0.0010 | 12.63 | 21.0201 | - |
| 950 | 0.0050 | 12.63 | 16.8403 | 16.6861 |
| 950 | 0.0100 | 12.63 | 15.8876 | 15.4176 |
| 950 | 0.0500 | 12.63 | 13.967 | 14.0240 |
| 950 | 0.3000 | 12.63 | 12.3852 | 12.2633 |
| 1000 | 0.0001 | 21.1 | 95 | 94.8702 |
| 1000 | 0.0002 | 21.1 | 76 | - |
| 1000 | 0.0007 | 21.1 | 52.4296 | 52.3033 |
| 1000 | 0.0020 | 21.1 | 37.7328 | 37.5001 |
| 1000 | 0.0100 | 21.1 | 27.9388 | 28.2090 |
| 1000 | 0.0500 | 21.1 | 23.9435 | 23.9700 |
| 1000 | 0.1000 | 21.1 | 22.6202 | 22.3469 |
| 1000 | 0.3000 | 21.1 | 21.3700 | 21.4938 |
| 1050 | 0.0001 | 77.57 | 164.8713 | 165.0638 |
| 1050 | 0.0100 | 77.57 | 46.0404 | 46.0867 |
| 1050 | 0.0200 | 77.57 | 40.4298 | 40.5404 |
| 1050 | 0.0500 | 77.57 | 37.4276 | 37.3475 |
| 1050 | 0.1000 | 77.57 | 36.2312 | 36.1864 |
| 1050 | 0.3000 | 77.57 | 33.9519 | - |
| 1100 | 0.0001 | 117.79 | 253.4429 | 253.4109 |
| 1100 | 0.0010 | 117.79 | 131.3201 | 131.3422 |
| 1100 | 0.0100 | 117.79 | 70.5940 | 70.5405 |
| 1100 | 0.0300 | 117.79 | 61.9280 | 61.9516 |
| 1100 | 0.1000 | 117.79 | 53.9414 | 53.9305 |
| 1100 | 0.3000 | 117.79 | 49.1312 | 49.1622 |

### 3.3.2. Construction of Neural Network Model to Obtain Grain Size

In the same way as in the previously developed ANN model, temperature and strain rate were normalized to values between 0 and 1. Grain size values do not need to be normalized. The ANN model is the same as the one previously developed used to determine the flow curves, that is, the Multi-Layer Perceptron (MLP) based on Feed-Forward and backpropagation (BP) as a learning algorithm. For training, the Levenberg–Marquardt (trainl) algorithm is applied, based on the backpropagation (BP) learning algorithm.

The network architecture we adopt is composed of three layers, input, test and validation. The input layer is composed of three elements ($T$, $\dot{\varepsilon}$ and $G_{s.INITIAL}$); two hidden layers with twelve and nine neurons; and as output layer, we will have the final grain size, recrystallized ($G_{s.FINAL}$). The architecture of the network, 3-11-6-1, is represented in Figure 12.

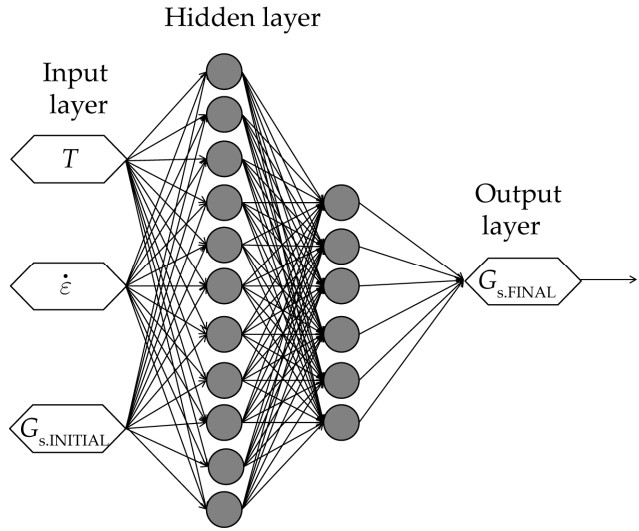

**Figure 12.** ANN to obtain final grain size.

To develop the ANN to predict the final grain size, few experimental data were used to train, validate and test the proposed ANN. In total, there were 33 experimental data, of which we used 28 (84.8%) for training and 5 (15.2%) for validation (Table 7). For testing, 8 data were chosen randomly from unseen data experiments (Table 8). It was proved, by using these data, that the trend of the evolution of the final grain sizes predicted by the ANN, as a function of temperature and strain rate, is very similar to the experimental trend. Therefore, predictive results were obtained without further experimental testing, thereby allowing us to reduce the amount of costly experimental testing. Finally, by comparing experimental data and ANN predicted results, it is accepted that both sets of data are very similar. So, the proposed ANN can be used reliably for predicting the final grain size obtained during the hot forming process. The best results were obtained for a minimum square error (MSE) of 0.000005. The training phase stopped after 807 iterations.

**Table 7.** Results of ANN validation phase.

| T (°C) | $\dot{\varepsilon}$ (s$^{-1}$) | $G_{s.FINAL}$ (μm) | *ANN* (μm) | $e_{rel}$ (%) |
|---|---|---|---|---|
| 900 | 0.01 | 11.3416 | 10.9527 | 3.43 |
| 950 | 0.0010 | 21.0201 | 21.9764 | 4.55 |
| 1000 | 0.0002 | 76 | 82.6071 | −8.69 |
| 1050 | 0.0500 | 23.9436 | 23.9700 | −0.11 |
| 1100 | 0.3000 | 33.9519 | 32.3338 | 4.76 |

**Table 8.** Results of ANN testing phase.

| T (°C) | $\dot{\varepsilon}$ (s$^{-1}$) | $G_{s.INITIAL}$ (μm) | $G_{s.FINAL}$ (μm) *ANN* |
|---|---|---|---|
| 900 | 0.0001 | 11.12 | 21.6098 |
| 900 | 0.001 | 11.12 | 12.8276 |
| 900 | 0.01 | 11.12 | 10.9527 |
| 900 | 0.1 | 11.12 | 9.8426 |
| 950 | 0.0001 | 12.63 | 3.4371 |
| 950 | 0.001 | 12.63 | 21.9764 |
| 950 | 0.01 | 12.63 | 15.4176 |
| 950 | 0.1 | 12.63 | 12.848 |

**Table 8.** *Cont.*

| T (°C) | $\dot{\varepsilon}$ (s$^{-1}$) | $G_{s.INITIAL}$ (μm) | $G_{s.FINAL}$ (μm) *ANN* |
|---|---|---|---|
| 1000 | 0.0001 | 21.1 | 94.8702 |
| 1000 | 0.001 | 21.1 | 45.5325 |
| 1000 | 0.01 | 21.1 | 28.209 |
| 1000 | 0.1 | 21.1 | 22.3469 |
| 1050 | 0.0001 | 77.57 | 165.0638 |
| 1050 | 0.001 | 77.57 | 83.6777 |
| 1050 | 0.01 | 77.57 | 46.0867 |
| 1050 | 0.1 | 77.57 | 36.1864 |
| 1100 | 0.0001 | 117.79 | 253.4109 |
| 1100 | 0.001 | 117.79 | 131.3422 |
| 1100 | 0.01 | 117.79 | 70.5405 |
| 1100 | 0.1 | 117.79 | 53.9305 |

### 3.3.3. ANN Results

The last column of Table 6 shows the results obtained by the ANN after the training phase. In this last column, the rows without values were used for the ANN validation phase and are presented in Table 7. The relative error was calculated by using Equation (8).

The results obtained by the ANN can be compared with the experimental results shown in Figures 13 and 14. In these figures, it is clearly observed that the range of high temperatures and very low strain rates exhibit grain growth, and as the strain rate increases, the grain is refined. In this case, the initial grain size is different at each temperature.

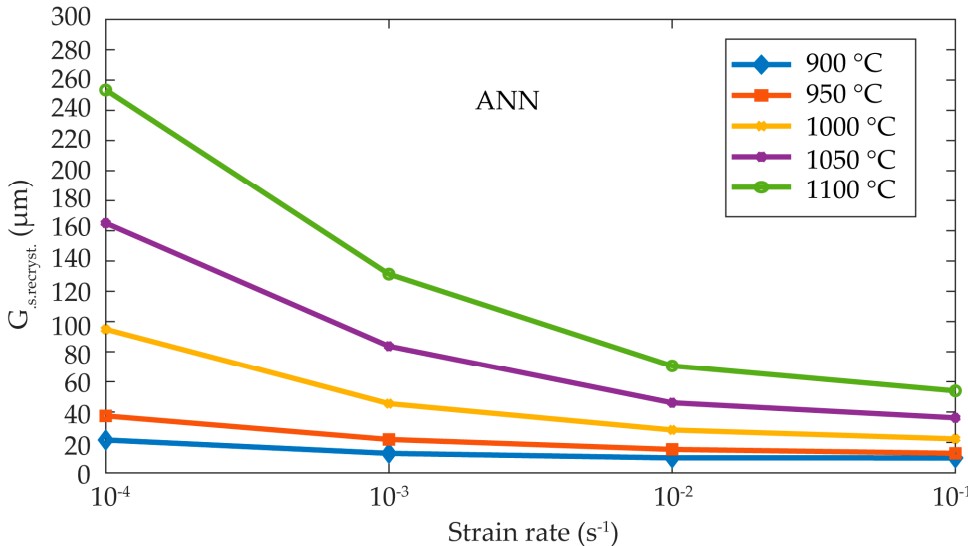

**Figure 13.** Evolution of recrystallized grain size for different strain rates.

Regardless of the theories that attempt to explain these processes through nucleation and growth mechanisms during DRX, the existence of grain refinement for single-peak recrystallization is evident.

The interpretation of the domains that appear in the processing maps of the studied microalloyed steel can be performed by using the flow curves associated with each domain, as well as the evolution of grain size with temperature and strain rate. Therefore, the broad domain, observed in the processing maps, located at low strain rates represents single-peak dynamic recrystallization in the temperature range between 900 °C and

950 °C, while at higher temperatures (950 °C), this same domain is representative of cyclic dynamic recrystallization. The domain located at high strain rates and high temperatures represents the process of single-peak dynamic recrystallization. The microstructure of the specimens deformed at $T = 1100$ °C and $\dot{\varepsilon} = 0.3$ s$^{-1}$, as represented in Figure 15 [19]. In these microstructures, it is noteworthy that considerable reconstitution of the microstructure due to DRX processes and perhaps also additional post-dynamic processes is evident.

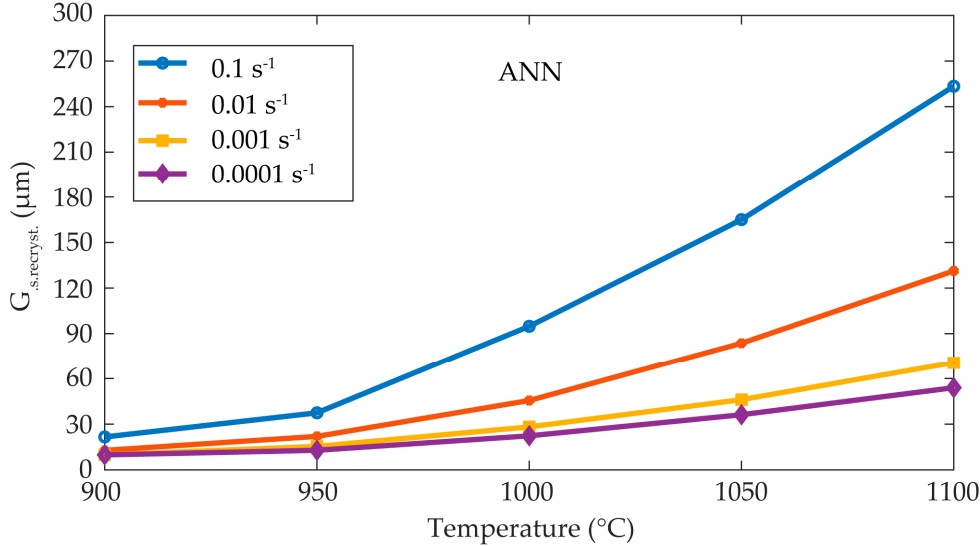

**Figure 14.** Evolution of recrystallized grain size for different temperatures and deformation rates.

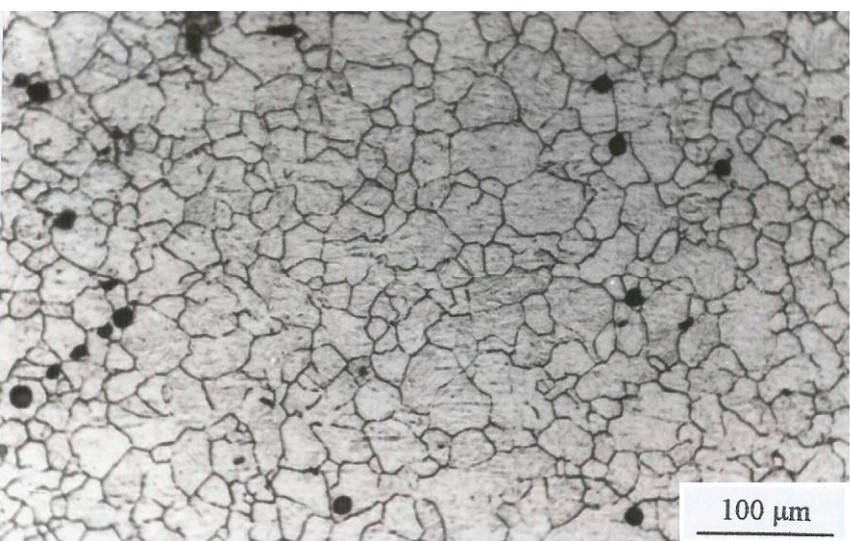

**Figure 15.** Final deformed microstructure of the studied steel at 0.3 s$^{-1}$ and 1100 °C.

## 4. Conclusions

This work focused on the study of dominant deformation mechanisms that control the microstructure, through grain size control, during the hot plastic deformation of a medium carbon microalloyed steel. A contribution of great interest and originality is the thermodynamic analysis of the deformation conditions that can lead to the appearance of plastic instabilities in the material.

Two artificial neural network (ANN) models were used successfully. The first was used to develop the processing maps for a medium carbon microalloyed steel subjected to a hot forming process. The second model is capable to determine the recrystallized grain size in a steady state. The ANN is trained by using temperature, strain, and strain rate as input

data and flow stress as target or output data. The network model used is the MLP (Multi-Layer Perceptron) and backpropagation (BP) learning algorithm. For network training, the Levenberg–Marquardt (trainlm) algorithm is applied, based on the backpropagation (BP) learning algorithm. The processing maps of the studied steel were developed based on the dynamic materials model (DMM) and its variant (VDMM). In these maps, it was possible to define safe areas for forming and areas to avoid. These results are of great industrial interest because they allow for choosing the most appropriate control parameters to carry out the forming process with full guarantees of success and reliability. The comparison between the experimental processing maps and those of the ANN show a very high similarity, which demonstrates the robustness and reliability of the ANN proposed in this work.

In relation to the ANN model proposed to determine the recrystallized grain size, the same method was used as in the first ANN, with the difference that, in this case, very few data were available to train and validate the network. However, the ANN results are very close to the experimental ones, so it is considered that the training, validation and testing were satisfactory. The graphs obtained in relation to the recrystallized grain size follow the trend of other studies carried out, so it can be confirmed that the results obtained by using the proposed ANN are correct.

Artificial neural networks are an efficient method to simulate the flow behavior of materials subjected to hot forming processes to determine the control parameters necessary to carry out safe forming without plastic instabilities. In addition, this method allows for predicting the evolution of the shaped microstructure through grain size.

The obtained results and the ability of the proposed ANN to predict safe domains and optimize process forming parameters facilitate hot forming process efficiency improvement through the modifications of the forming sequences and the heating strategies. Also, the capability of the proposed ANN of predicting flow curves for unseen deformation conditions can significantly ease the path for future research works to develop a promising numerical computing tool in designing hot forming processes for other medium carbon microalloyed steels with different chemical compositions without need to carry out experimental tests that are time- and cost-consuming. At an industrial level, this is particularly interesting for processes that involve multiple stages of deformation.

Finally, to have a deeper understanding of the control mechanisms of the complex flow behavior of medium carbon microalloyed steel, more microstructural research is needed, especially transmission and scanning electron microscopy and EBSD analysis to identify the dominant deformation mechanisms.

**Author Contributions:** Conceptualization, A.A.O., P.C. and J.I.A.; Data curation, J.I.A.; Formal analysis, P.C., J.I.A. and E.P.; Funding acquisition, A.A.O. and E.P.; Investigation, A.A.O., P.C. and J.I.A.; Methodology, A.A.O. and J.I.A.; Software, J.I.A.; Supervision, A.A.O. and P.C; Validation, A.A.O. and J.I.A.; Writing—original draft, A.A.O., P.C., J.I.A. and E.P.; Writing—review and editing, A.A.O., P.C., J.I.A. and E.P. All authors have read and agreed to the published version of the manuscript.

**Funding:** This research was funded by CICYT (Spain), through the research competitive project under grant number PID2020-114819GB-I00.

**Data Availability Statement:** The original contributions presented in the study are included in the [19], further inquiries can be directed to the corresponding author.

**Conflicts of Interest:** The authors declare no conflicts of interest.

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
