# Peer review of "Development of Neural Networks to Study Flow Behavior of Medium Carbon Microalloyed Steel during Hot Forming"

_metals, doi:10.3390/met14050554_

Round 1

Reviewer 1 Report

Comments and Suggestions for Authors

1.     The literature review is weak: authors must strengthen the introduction section by introducing the need of ANN in developing the plastic instability maps of a medium carbon micro alloyed steel during hot forming process.

2.     There are many artificial neural networks architectures are available and the selection of best architecture needs justification with supporting literature.

3.     The group citations must be avoided. Each individual contribution must be explained in the literature.

4.     The reason for fixing the values of steel austenite for 30 min at test temperatures of 900 oC, 1050 oC and 1150 oC need justification.

5.     The images presented in Figure 1 must be compared with identical scale bar.

6.     The authors used normalization between 0 and 1, however the equation 1 used for that purpose need supporting literature.

7.     The reason for selection of log-sigmoid transfer function need justification.

8.     How did the architecture selected and optimized as 3-12-9-1. Why the single layer hidden network was not tested and how did the number of hidden neurons were fixed.

9.     How did the grain size measured need to explained in the revised literature.

10.  Why did the author split to two neural network architecture for predictions. Instead they can predict the multiple outputs simultaneously.

Reviewer 2 Report

Comments and Suggestions for Authors

This paper studies the use of Artificial Neural Networks (ANN) in modeling plastic instability and recrystallized grain size for medium carbon steels in the hot forming process. ANN-models are based on experimental data.

Introduction. The description of the network training is quite clumsy. It seems that the authors have tried to avoid using the same words that thousands of papers have used before, and it has led to quite strange expressions. Especially, the second phase (testing) is badly formulated: What is the “test pattern” (input-output pairs)? What are the “final features”(network outputs)? What are the “values of the neurons in the last layer” (network outputs or weights)? It would be good to say at the very start that the data is divided into three parts: training, testing and validation data. The authors seem to mix the testing and validation phases. In validation, the trained network is usually used with unseen data from the same problem, and its performance is checked. A new problem requires new training. If the phases here  are to describe the application of ANN, a phase is missing: the use of the network model.

Introduction misses the problem definition, the list of contributions and the claims for the novelty of this work.

Section 2. Are the numbers before Table 3 correct? “for testing 1296 (30%)”.

Section 3. The percentage errors in Table 7 are wrong; e.g. in the first case, the error is 15% instead of 1.5%. It is difficult to see how the results in Table 8 validate the model. In this case, the data is far from enough. There are only 32 data pairs and a two-layer network with at least the duplicate number of weighting factors in the model.

Conclusions. It is stated that in the second case there was a limited number of data for model validation. Actually, there was no data for validation at all. In this sense, there is no proof to assume that the results are correct. The conclusion also claims that ANN  are used in determining the control parameters of the hot forming process. Is this claim supported by the results? Few sentences concerning the future research potential is also needed.

Comments on the Quality of English Language

No bigger mistakes.

Reviewer 3 Report

Comments and Suggestions for Authors

    The authors investigated the application of an artificial neural network (ANN) model concerned of the development of plastic instability maps of a medium carbon micro-alloyed steel during hot forming process. Moreover, they proceed to create another neural network capable of providing the recrystallized grain size in the steady state resulting from forming deformation.

  The topic original or relevant in the research field. The conclusions consistent with the evidence and arguments presented in this article. The references are appropriate.

Comments 

1. You should describe what does it add to the subject area compared with other published material in detail in the chapter of conclusion.

2. Figure 1 and Figure 15 : Are these figures are reuse of reference 15 ? If you reuse these figures, you must get permissions of reuse from the university concerned, and you must write “Data were obtained from Ref. [15].” in the captions of the figures.

3. L159-160  Equation (3) : Show the original paper which this equation was used for.

4. Figure 3 (b) : The dispersion from the straight line approximation of the data point of the lower right is large. Describe this occasion.

5. Figure 5 (d) data at 950 oC : Describe the reason of the deviation of the experimental result from the ANN.

Comments on the Quality of English Language

Minor editing of English language required. 

For example : L362 "... and validate the network. But (---> However) , despite this, ..."

Translate it into written language, not the spoken language.

Reviewer 4 Report

Comments and Suggestions for Authors

In this manuscript, the authors present the application of an artificial neural network (ANN) model to analyze the flow behavior of a medium carbon microalloyed steel during hot forming and make a comparison between ANN and experimental maps. This ANN method allows for predicting the evolution of the shaped microstructure, through the grain size. In this study they used two artificial neural networks (ANN) models: (i) one of them to develop the processing maps for a medium carbon microalloy steel, subjected to a hot forming process, and the other (ii) to determine the recrystallized grain size in steady state. The ANN is trained using temperature, strain, and strain rate as input data and flow stress as target or output data.

For this purpose, the authors apply the dynamic materials model (DMM) and a variant of the DMM (VDMM) according to which the safe hot-forming domains and the plastic instability domains of the studied material are delineated. They chose these two methods to obtain the optimal hot forming zones and, consequently, safe parts without microstructural defects. They analysed the microstructure modifications (the evolution of recrystallized grain size for different strain rates) during hot compression tests.

In a hot forming process, both dynamic recrystallization (DRX) and dynamic recovery (DRV) are considered beneficial mechanisms for the microstructure and consequently, the mechanical properties of the formed material, as they provide stable flow and improve the formability of the material.

They concluded that, based on results obtained by ANN compared with the experimental results the range of high temperatures and very low strain rates exhibit grain growth, and as the strain rate increases the grain is refined.

In my opinion, the paper is well written and the results of the manuscript are of interest to researchers in computational material science because they allow choosing the most appropriate control parameters to carry out the forming process with success and reliability.

 As a result, I agree with the publication of this paper, after a minor revision.

Previous to the publication, the following minor details should be corrected:

Please develop in detail the experimental method used to obtain the experimental data. It is not sufficient to refer to the bibliographic reference [15].

Round 2

Reviewer 2 Report

Comments and Suggestions for Authors

This is the revised version of the paper that I reviewed in the earlier stage. I see that the authors have taken my comments into account and the paper has improved. I can now recommend its publication.